# The Impact of Extracellular Histones and Absence of Toll-like Receptors on Cardiac Functional and Electrical Disturbances in Mouse Hearts

**DOI:** 10.3390/ijms25168653

**Published:** 2024-08-08

**Authors:** Randall Loaiza, Fatemeh Fattahi, Miriam Kalbitz, Jamison J. Grailer, Mark W. Russell, Jose Jalife, Hector H. Valdivia, Firas S. Zetoune, Peter A. Ward

**Affiliations:** 1Center for Arrhythmia Research, University of Michigan, Ann Arbor, MI 48109, USA; r.loaiza@gmail.com (R.L.); jjalife@umich.edu (J.J.); hhvaldiv@medicine.wisc.edu (H.H.V.); 2CENIBiot Laboratory, The National Center of High Technology (CeNAT-CONARE), San José 10109, Costa Rica; 3Division of Allergy and Clinical Immunology, Department of Internal Medicine, University of Michigan Medical School, Ann Arbor, MI 48109, USA; ffattahi@umich.edu; 4Department of Pathology, University of Michigan Medical School, Ann Arbor, MI 48109, USA; m.kalbitz@googlemail.com (M.K.); jamison.grailer@promega.com (J.J.G.); zetoune@umich.edu (F.S.Z.); 5Department of Orthopaedic Trauma, Hand, Plastic and Reconstructive Surgery, University Hospital of Ulm, 89081 Ulm, Germany; 6Military Medical City Hospital, Doha 486441, Qatar; 7Integrated Biology R&D, Bioassay Development, Promega Corporation, Madison, WI 53711, USA; 8Department of Pediatrics, University of Michigan Medical School, Ann Arbor, MI 48109, USA; mruss@umich.edu; 9Centro Nacional de Investigaciones Cardiovasculares (CNIC), 28029 Madrid, Spain; 10Department of Medicine, Cardiovascular Research Center, University of Wisconsin School of Medicine and Public Health, Madison, WI 53705, USA

**Keywords:** complement, C5a, sepsis, heart dysfunction, arrhythmia, cardiomyocytes, neutrophils

## Abstract

In polymicrobial sepsis, the extracellular histones, mainly released from activated neutrophils, significantly contribute to cardiac dysfunction (septic cardiomyopathy), as demonstrated in our previous studies using Echo-Doppler measurements. This study aims to elucidate the roles of extracellular histones and their interactions with Toll-like receptors (TLRs) in cardiac dysfunction. Through ex vivo assessments of ECG, left ventricle (LV) function parameters, and in vivo Echo-Doppler studies in mice perfused with extracellular histones, we aim to provide comprehensive insights into the mechanisms underlying sepsis-induced cardiac dysfunction. Langendorff-perfused hearts from both wild-type and TLR2, TLR3, or TLR4 knockout (KO) mice were examined. Paced mouse hearts were perfused with histones to assess contractility and relaxation. Echo-Doppler studies evaluated cardiac dysfunction after intravenous histone injection. Histone perfusion caused defects in contractility and relaxation, with TLR2 and TLR3 KO mice being partially protected. Specifically, TLR2 KO mice exhibited the greatest reduction in Echo-Doppler abnormalities, while TLR4 KO exacerbated cardiac dysfunction. Among individual histones, H1 induced the most pronounced abnormalities in cardiac function, apoptosis of cardiomyocytes, and LDH release. Our data highlight significant interactions between histones and TLRs, providing insights into histones especially H1 as potential therapeutic targets for septic cardiomyopathy. Further studies are needed to explore specific histone–TLR interactions and their mechanisms.

## 1. Introduction

One of the critical organ systems profoundly impacted during severe sepsis is the cardiovascular system, notably the heart. Organ dysfunction arising from sepsis has been extensively researched [1]. Cardiac dysfunction emerges as a frequent complication in septic patients and septic mice, often culminating in a substantial mortality rate. Sepsis in humans is frequently associated with mortality rates ranging from 70% to 90% with cardiovascular impairment compared to 20% in those without impairment, highlighting the critical role of cardiovascular dysfunction in sepsis-related mortality [2]. These dysfunctions are distinct from ischemia-reperfusion injury and are reversible if the patient survives sepsis [1]. Surprisingly, in humans with infectious sepsis, coronary arterial blood flow measurements have demonstrated unimpeded flow and normal tissue pO2 levels [3,4].

More than two decades ago, evidence suggested depressed myocardial performance during sepsis, indicating a role of factors contributing to cardiac dysfunction [5]. Significant elevations in serum cytokines, such as IL-1β, IL-6, TNF-α, IL-8, and the IL-7 family of cytokines, have been noted during sepsis [6,7,8,9,10]. The inflammatory responses initiated by these cytokines likely contribute to widespread organ system failure in sepsis. We have shown that sepsis in mice activates the NLRP3 inflammasome, leading to the release of IL-1β, IL-6, and other cytokines and chemokines, causing cardiomyocyte (CM) dysfunction [11,12]. Subsequent research has highlighted the role of extracellular histones, released into the extracellular compartment during sepsis, as an important driver inducing myocardial dysfunction during sepsis [13]. These histones are the chief component of chromatin that can act as damage-associated molecular patterns (DAMPs) when released to the extracellular compartment and have been postulated to be a major cause of cardiac failure along with other organ failures during sepsis [11,14,15,16,17]. Both C5a receptors (C5aR1 and C5aR2) are known to be required for histone appearance in plasma during polymicrobial sepsis [11,18,19]. Histones are likely released from neutrophil extracellular traps (NETs) and macrophage extracellular traps (METs) after activation of phagocytes by C5a at the acute phase of sepsis [11,20]. The histones are postulated to play a significant role not only in sepsis but also in trauma-related organ dysfunction [21,22] and non-infectious conditions that clinically mimic sepsis such as chemical injury or ischemic injury [23,24]. In humans with infectious sepsis, plasma histones (H2B, H3) have been found at levels similar to those observed in septic mice, further emphasizing their potential significance [25]. Notably, deaths occurring during septic shock are often linked to cardiac dysfunction caused by extracellular histones [13]. Using a neutralizing antibody to histones, immunofluorescence studies revealed histones appearance in CMs and in plasma after cecal ligation and puncture (CLP), involving heart dysfunction [11]. Additionally, observations in COVID-19 infections have suggested evidence of extracellular histone release and CM dysfunction, further highlighting the relevance of our findings [26,27]. Our current study provides evidence that septic cardiac abnormalities occur due to histones.

Electrical and mechanical dysfunctions in the heart often develop in septic patients and are frequently associated with a poor prognosis. Ventricular arrhythmias are a serious complication of sepsis, often associated with increased mortality risk in ICU-admitted patients [28]. Other cardiac complications include decreased ventricular contraction, defective relaxation kinetics, ventricular dilatation, and electrical conduction abnormalities [29,30,31]. Septic patients may exhibit prolongation of QRS complexes, as defined by electrocardiography (ECG), which usually disappears during recovery from septic shock, indicating reversibility [32]. In the septic mice, these disturbances in functional responses of hearts were defined by Echo-Doppler parameters [11]. In polymicrobial sepsis in mice, early cardiac defects in CM functions have been linked to reduced levels of homeostatic proteins, notably Na+/K+-ATPase [33]. Prolongation of the action potentials is another CM defect observed during sepsis [33]. Developing and validating a clinical risk score for QT prolongation has allowed for the prediction of which septic patients are at the highest risk for developing sepsis-associated cardiomyopathy, aiding in monitoring and treatment decisions [34,35].

Toll-like receptors (TLRs) notably TLR2 and TLR4, are proposed as receptors for extracellular histones [14,23]. Our previous research also suggests TLR3 and TLR9 as other potential histone receptors, supported by echocardiographic studies showing the protective effect in the absence of these TLRs [36]. As stated above, extracellular histones serve as DAMPs, released mainly from NETs during sepsis. Increasing evidence links TLR activation, especially TLR2, TLR3, and TLR4, to responses observed during septic cardiomyopathy in mice [37]. Histones may bind to several TLRs, suggesting their potential role in cardiac dysfunction [23,24,36,38]. While it is established that sepsis causes cardiac dysfunction [1,11,29,39,40], the specific contribution of extracellular histones to cardiac dysfunction and ECG changes needs further investigation. The current study design investigates the direct effects of histones on cardiac function and assesses the impact of various TLRs, including TLR2, TLR3, and TLR4 knockout (KO) on changes in electrical and functional responses in the mouse heart ex vivo and in vivo. Our study also directly examines the effects of individual histones (H1, H2A, H2B, H3, H4) on cardiac function and the defective electrical responses of hearts exposed to these individual histones. We explore the role of each individual histone on perfused hearts concerning electrical and functional activities in hearts from wild-type (Wt) mice. These investigations aim to identify histones that are pathogenic for CMs while considering the impact of different TLRs and each individual histone on cardiac abnormalities.

## 2. Results

### 2.1. Functional Abnormalities in Wt Mouse Hearts Perfused Ex Vivo with Histone Mix

For these studies, Wt mouse hearts electrically paced were perfused for up to 20 min at 37 °C with control buffer (PBS), with the histone mix (Figure 1A–F). For each condition, the concentration of the histone mix varied from 5–20 µg/mL as indicated and involved the use of 11 hearts for each condition. As indicated, several different functional parameters were measured. In Figure 1A, either buffer or the histone mix (5, 10, or 20 µg/mL) was used as the perfusate. Tracings for both the buffer and different doses of histone mix are shown in the expanded view on the left portion of the tracing for 300 milliseconds (ms), while the right side (black compact tracings) represents 15 s of perfusion. Perfusion of hearts with buffer resulted in tracings shown in the first left panel of Figure 1A. Subsequent perfusion with three different concentrations of the histone mix (5, 10, 20 µg/mL) in the remaining panels showed progressively reduced Left Ventricular (LV) Pressures (LVP) amplitudes, with reductions and widening of the amplitudes. At the 20 µg/mL concentration of the histone mix, LVP amplitudes collapsed, and bradycardia developed. In Figure 1B, the temporal progression is depicted following the perfusion of a histone mix (5 µg/mL), as shown in the expanded view in Figure 1C. Within 1 min of perfusion, observable widening of amplitudes and the onset of bradycardia became apparent. By the 8th minute, LVP had collapsed, and severe bradycardia had developed. In Figure 1D–F, perfused Wt hearts were assessed for LVP (Figure 1D), end-diastolic pressures (Figure 1E), or relaxation pressures (Figure 1F) over a 20 min perfusion time. The horizontal dashed lines indicate the point of heart failure, where tracings revealed ≥80% loss of pressure values, denoting heart failure. Black bars represent responses to buffer perfusion (Wt Control), while white bars represent pressures in hearts perfused with the histone mix (Wt histones) at different doses (5–20 µg/mL). As evident in Figure 1D, LVP and end-diastolic pressures (Figure 1E) began to decline with the perfusion containing 10 µg/mL of the histone mix. With the histone mix of 20 µg/mL, there were statistically significant reductions in all three pressure frames (Figure 1D–F). In Figure 1F, relaxation pressures were measured. Starting with the histone mix of 5 µg/mL, there was a decline in relaxation pressures, which became more pronounced as the histone mix concentrations of 10 µg/mL and 20 µg/mL were used.

### 2.2. ECG Changes in Wt Hearts Perfused with Histones or After Induction of Polymicrobial Sepsis by CLP

In Figure 2A, Wt hearts (*n* ≥ 5 hearts for each experiment) were perfused with three different concentrations of the histone mix (5, 10, or 20 µg/mL) for up to 12 min at 37 °C. By 12 min of perfusion there was a distinct widening of the QT interval (Figure 2A). The baseline duration of the QTc interval of the heart was 67.4 ± 3.1 ms, which then increased to 71.8 ± 3.6 ms, and 87.2 ± 5 ms only 6 and 12 min after histone perfusion, respectively (Figure 2A). At the two higher concentrations of the histone mix, (10, 20 µg/mL) there was collapse of ECG tracings. In Figure 2B, hearts were perfused with the histone mix (5 µg/mL) mix over a 12 min period of time. Figure 2B-1 shows a significantly increased QT interval by 6 min of perfusion (approximately 15 ms increase), peaking at 8 min (22 ms increase in QT interval), followed by a gradual decline (all changes became significant at 6 min and remained so through 12 min). In Figure 2B-2, there was a steady increase in the QRS interval starting at 6 min of perfusion and continuing at 12 min. In Figure 2B-3, there was no increase in the PR interval. ECG recordings were also obtained in Wt mice before CLP (time 0) and 6 and 12 h after polymicrobial sepsis (Figure 2C). Sham controls and CLP mice (n = 5 mice in each group) were used 6 and 12 h after CLP. The following intervals (ms) were measured: QT, QRS, and PR. The QT intervals in the septic mice were significantly elevated at 6 hr and further elevated at 12 after CLP. The QRS interval was also significantly elevated at 6 and 12 h in CLP hearts. Consistent with the studies of hearts perfused with histone mix (Figure 2B), sepsis did not cause statistically significant changes in the PR interval (Figure 2C).

### 2.3. Ability of Histone Mix to Cause Cessation of Beating Mouse Hearts as a Function of Histone Concentration and Effect of TLR Absence

As shown in Figure 3A, Wt non-paced spontaneously beating mouse hearts were perfused with buffer or with the histone mix at increasing concentrations (5–20 µg/mL). The cessation of beating hearts was determined by visual inspection. In the presence of buffer, all hearts over 12 min of exposure (n ≥ 10 hearts) per group continued beating during perfusion with buffer. In the presence of 5 µg/mL of the histone mix, 63% of the hearts were still beating after 12 min. As the concentrations of the histone mix were increased to 10, 15, and 20 µg/mL, there was a progressive increase in hearts caused by progressively reduced beating or cessation of beating. At the histone mix of 20 µg/mL, all 11 hearts had ceased beating over the interval of 12 min. Figure 3B contrasts these responses with non-paced spontaneously beating mouse hearts from various TLR KOs including TLR2, TLR3, and TLR4 KO compared to Wt, as described in Figure 3A. With 10 µg/mL of the histone mix, the percentage of functional (beating) Wt hearts decreased to 20% in Wt hearts and this number decreased to 0% in the presence of 20 µg/mL histones in Wt or TLR4 KO hearts. At all concentrations, the detrimental effects of the histones were evident 5 min after the start of perfusion. TLR2 KO hearts show more protection from the adverse effects of the histone mix. LV dysfunction (reduced contractility) occurred in only one out of eleven hearts perfused with 5 µg/mL histones, while in 80%, the function was maintained. TLR3 KO hearts were also more resistant to injury, with nearly 30% of TLR3 KO hearts still beating 20 min after perfusion with the highest concentration of histones. In contrast, the absence of TLR4 provided no protection for the beating hearts at any concentration of the histone mix. Accordingly, the absence of TLR4 surprisingly did not afford any protective effects during ex vivo perfusion with histones, in marked contrast to the protective effects in TLR2 and TLR3 KO hearts. These findings highlight the protective effects of TLR2 KO, and to a lesser extent TLR3 KO, in maintaining heart function during histone mix perfusion, while such protective effects were absent in TLR4 KO hearts.

### 2.4. Functional Abnormalities in Wt Mouse Hearts Perfused with Histone Mix and Effects of TLR Absence on Functional Responses of Mouse Hearts to Histone Mix

In our investigation of functional abnormalities, electrically paced Wt mouse hearts were compared with hearts from TLR2, TLR3, and TLR4 KO mice, all perfused for up to 20 min at 37 °C with buffer or the histone mix. The mechanical function was evaluated in Langendorff-perfused hearts (n = 11 Wt and n ≥ 8 for each TLR KO mice). Perfusing hearts with three different concentrations of the histone mix (5, 10, 20 µg/mL) led to progressive reductions in LVP generation, characterized by both reduced and widened LVP amplitudes in Wt hearts, as also described in Figure 4A. At the 20 µg/mL concentration of the histone mix, LVP amplitudes in Wt hearts plummeted to the point of collapse, accompanied by the development of bradycardia.

As shown in Figure 4B (n = 8 hearts), baseline (ctrl buffer perfused) LVP were similar in TLR2, TLR3, and TLR4 KO hearts, with pressures reaching nearly 100 mmHg. In TLR2 KO hearts, the areas between the LV pressures began to widen with the two concentrations of histones. In TLR3 KO hearts, there was a slight (20%) drop in pressures at the higher concentration (20 µg/mL) of histones. In the case of TLR4 KO hearts, the responses were dramatically different. At both concentrations of histones (10, 20 µg/mL), the LV pressures were virtually ablated. The reasons why LVP was abolished in hearts from TLR4 KO mice exposed to the histone mix as exposed to responses of TLR2 and 3 KO hearts are not currently known.

In summary, TLR2 KO hearts demonstrated a protective effect, exhibiting minimal changes in LV function exposed to histone mix doses (20 µg/mL). TLR3 KO hearts, to a lesser extent, showed protection with changes in LV function initiating at the 20 µg/mL dose. In stark contrast, TLR4 KO hearts displayed no protection with LV function changes evident at a lower histone mix dose (10 µg/mL), indicating a lack of resilience compared to the Wt mouse heart. These findings highlight the distinct protective effects of TLR2 KO and, to a lesser extent, TLR3 KO while underscoring the vulnerability of TLR4 KO hearts in the face of histone mix-induced LV functional alterations.

### 2.5. Effect of TLR Absence on Delta QTc Prolongation in Murine Hearts Perfused with Histone Mix

In Figure 5, Wt paced hearts were perfused with buffer (baseline) or the histone mix (5, 10, and 20 µg/mL), and ECG recordings were measured for 12 min with 2 min intervals. Due to marked abnormalities in hearts perfused with 10 or 20 µg/mL of the histone mix, the focus was placed on changes in hearts exposed to the 5 µg/mL histone mix at the 12 min perfusion interval. At this point, the QT interval (QTc) was clearly prolonged, as described in Figure 5A. Figure 5A demonstrates the ECG results over 12 min, representing hearts perfused with the histone mix (5 µg/mL), revealing marked and rapid changes in QTc over time and genotypes. Wt hearts displayed progressive QTc prolongation during the first 8 min, followed by a plateau. Concurrently, QTc interval of TLR2 and TLR3 KO mice remained consistent with the baseline (Figure 5A). In sharp contrast, TLR4 KO hearts displayed the largest average increase in the QTc interval among the three groups (Figure 5A). Thus, TLR2 and TLR3 KO mice showed minimal changes, suggesting protection. In contrast, TLR4 KO mice showed exaggerated QTc prolongation at all time points, accentuating the effects of histone mix exposure. Figure 5B compares the ECG pattern of TLR4 KO (worse) and TLR2 KO (protective), evidently showing that the most significant variations in ECG were observed in the duration of the QTc interval.

Comparatively, the figure with detailed values showcased the impact of TLR absence on corrected QTc duration. Wt mice exhibited increasing ΔQTc values over time, reaching 18.5 ms at 12 min. TLR2 KO mice showed a protective effect with significantly lower ΔQTc values (slightly increased by 2 ms) while TLR3 KO mice displayed reduced prolongation (increased by 9 ms), indicating robust protection at 12 min. Conversely, TLR4 KO mice exhibited significantly higher ΔQTc values (increased by almost 40 ms from 68 ± 5 ms at baseline only 8 min after histones perfusion), suggesting a pro-arrhythmic effect. In summary, TLR2 and TLR3 KO provided protection against QTc prolongation, with TLR2 KO showing the most significant reduction. In contrast, TLR4 KO led to pronounced pro-arrhythmic effects, highlighting the dynamic responses of murine hearts to TLR KO.

### 2.6. Echo-Doppler Parameters Measurement after Intravenous Infusion of Histone Mix

As a follow-up to the data in previous figures, which indicated that the absence of TLR2 or TLR3 protected the heart from defects developing after ex vivo perfusion of hearts with the histone mix, while the absence of TLR4 caused an intensification of functional defects, complementary in vivo experiments were conducted in Wt, TLR2 KO (as the most protective), and TLR4 KO (as the least protective) mice in terms of heart electrical and functional effects following histone mix perfusion. Mice were intravenously injected with the histone mix (65 mg/kg body weight), simulating levels of plasma histones appearing after CLP [11]. This amount of histone mixes employed mimics the plasma levels of histones appearing in the plasma of Wt mice after CLP [11]. As evident in Figure 6, both systolic and diastolic parameters were evaluated in Wt, TLR2 KO, and TLR4 KO mice at various time points (including 10, 60, 90 min, and 24 h as indicated) after i.v. histones injection. In Wt mice, a significant decrease in heart rate was noted 90 min after histones injection (Figure 6A) compared to values before injection. By 24 h, values had returned to the pre-infusion (time 0) values. A more intense drop in heart rate after histones infusion was observed in TLR4 KO but not in TLR2 KO animals during the first 90 min post-infusion. Stroke volumes also declined acutely after histones infusion in Wt mice (Figure 6B). However, stroke volumes were not significantly affected by histones infusion in TLR2 KO nor TLR4 KO mice. The lower heart rate and stroke volume resulted in a lower calculated cardiac output in Wt and TLR4 KO mice but not in TLR2 KO mice after histone infusion compared to pre-infusion values (Figure 6C). Heart rate, stroke volume, and cardiac output had essentially returned to baseline in all animals by the 24 h time point. Left ventricular ejection fraction was slightly increased in Wt and TLR4 KO animals after histone infusion (Figure 6D). This may represent an adaptation to loading conditions and heart rate rather than a primary effect of histone infusion to increase contractility. Again, TLR2 KO mice did not show a significantly sustained change in ejection fraction after histone infusion. There was also a modest reduction in other parameters (Figure 6E–H) after histone infusion into Wt animals, and to a greater extent in TLR4 KO animals. Isovolumic relaxation time was significantly prolonged after histone infusion into Wt and TLR4 KO mice, but this was not found in TLR2 KO mice (Figure 6E). The effect of histone infusion on other measures, including the mitral E/A ratio, was more variable, potentially due to differences in preload and secondary compensatory mechanisms between the pre- and post-infusion states (Figure 6F–H). Together, these data indicate an intensification of defective Echo-Doppler parameters in TLR4 KO mice compared to Wt mice and nearly no defective parameters developing in TLR2 KO mice following i.v. injection of the histone mix.

### 2.7. Effects of Individual Histones on LV Function, ECG Changes, and Cytokine Release on Perfused Hearts

Since we found the effect of histone mix in our previous figures, we extended our studies to elucidate the effect of each individual histone (H1, H2A, H2B, H3, and H4) present in the histone mix. Our earlier in vivo and in vitro studies showed the various biological effects of different individual histones [41]. For these studies, we used only the Wt hearts. The studies described in Figure 1 and Figure 2 were carried out similar to histone mix protocols.

In Figure 7A, hearts were perfused with buffer or individual histones (5 µg/mL) over a time frame of 0–150 min. H1 reduced LVP most significantly followed closely by H2A and H2B. H4 had little effect on LVP, while H3 induced slightly increased pressures above those pressures. When the perfusate was buffer, minimal effects developed. Based on these data, it is not surprising that H1 followed by H2A and H2B had the most inhibitory effects. As will be shown in Figure 8, these histones induced the most pronounced indications of CM lysis [lactate dehydrogenase (LDH) release] and apoptosis (TUNEL staining), techniques previously detailed [18,36,41].

In Figure 7B, we calculated the extent to which individual histones would alter QT and QRS intervals using perfusion of Wt hearts with each 5 of the individual histones (5 µg/mL). For each bar, n ≥ 5 mice. In Figure 7B-1, the QT interval was prolonged by perfusion with H1 by approximately 33 ms while H2A and H2B caused some prolongation (18–22 ms) of this interval. However, H3 and H4 did not cause prolongation of this interval. When the effects of individual histones on ECG parameters were examined, the same three histones (H1, H2A, and H2B) caused prolongation of the QRS interval (Figure 7B-2). In summary, these data indicate H1 as the most predominant individual histone causing functional and electrical dysfunction in the perfused hearts compared to other individual histones.

Figure 7C,D show the levels of cytokine release from Wt hearts perfused with individual histones. It is known that polymicrobial sepsis is often associated with cytokine release from various organs as well as from phagocytic cells [42,43,44,45,46,47,48,49]. Wt hearts were perfused with buffer or with individual histones (each at 5 µg/mL) for the times indicated (6–150 min), at 37 °C. Cytokine levels were measured in perfusion fluids by ELISA. Generally, IL-6, H1, and H2B caused the highest levels of cytokine release followed by H2A, H3, H4 (Figure 7C). In the case of TNF-α, the most robust responses occurred with H2B > H4 > H3, followed by H2A and H1 (Figure 7D). Different cell types in the heart where TNF-α and IL-6 originate might explain the difference. Interestingly, concentrations of TNF-α and IL-6 do not directly correlate with cardiac dysfunction or cell death. This observation suggests a complex pathophysiology that extends beyond simple cytokine release. Factors such as exposure duration, cytokine concentration, and model dependency must be considered, as evidenced by the contradictory effects of TNF-α observed post-myocardial infarction and in heart failure cases in humans and non-human models [50].

### 2.8. LDH Release and TUNEL Staining of Wt Hearts after Perfusion with Individual Histones

For the experiments in Figure 8A, Wt mouse hearts were perfused with buffer or with individual histones (5 µg/mL) for the indicated periods of time at 37 °C. The dose of individual histones (5 µg/mL) was the same as employed for most of the other experiments. As shown in Figure 8A, between 30 and 150 min of perfusion with H1, approximately 40% of LDH release occurred, virtually all of it during the first 30 min of perfusion, followed by a plateau, whereas, with the presence of the other individual histones, only 2–5% LDH release occurred.

To determine which individual histone causes the highest apoptosis in the heart LV, the TUNEL staining was employed as described before [18]. Figure 8B shows the TUNEL staining of LV frozen sections of perfused hearts with individual histones while Figure 8C shows the quantitative of the TUNEL staining. In accordance with the release of LDH from the perfused heart (Figure 8A), TUNEL staining of LV frozen sections of perfused hearts showed maximal TUNEL staining (approximately 20%) in hearts perfused with H1. Perfusion with the other histones resulted in little or no evidence of TUNEL-positive cells (Figure 8B). H2B and H4 showed no staining. In summary, these data indicate that H1 as the most predominant individual histones causing apoptosis in the CMs compared to other individual histones.

## 3. Discussion

Extracellular histones are key emerging proinflammatory and prothrombic factors, particularly in polymicrobial sepsis [14,16,19,51]. Histone release, predominantly from PMNs via complement activation products such as the C5a anaphylatoxin, leads to the formation of NETs, facilitating bacterial killing. Those NETs become the main source of circulating plasma histones in the polymicrobial sepsis induced by CLP [11]. Our investigations indicated that histones are linked to adverse outcomes in sepsis, including organ injury and early development of cardiac dysfunction (septic cardiomyopathy) [11]. Specifically, histone-blocking antibodies decreased the cardiac dysfunction in septic mice induced by CLP. Interfering with the C5a and NLRP3 inflammasome complex can mitigate histone appearance in polymicrobial sepsis models [11,12,33]. Accordingly, a recent study has confirmed the presence of plasma histones (H2B and H3) in sepsis patients, with concentrations reaching 10–20 µg/mL (as measured by ELISA) which closely align with the levels we observed in the plasma of septic mice [11].

In this report, we present findings that direct perfusion of a histone mixture into mouse hearts leads to functional and electrical abnormalities. Our data also provide evidence that some, but not all, individual histones have adverse effects on the heart, with H1 showing the largest overall alterations relative to the other individual histones. H1 can induce cell death (as shown by TUNEL staining and LDH) of LV CMs, similar to acute lung injury and phagocytes from earlier studies [41]. Also, certain histones can cause proinflammatory cytokines release from the heart, some of which have cardio-suppressive effects while others induce proinflammatory responses potentially exacerbating cardiac dysfunction and leading to worse outcomes [12,50]. Quite similar events occur in neutrophils and macrophages [41].

Although the molecular targets of histones have not been completely elucidated, their interaction with TLRs has attracted considerable attention [23,24,36,38]. Our findings from TLR KO hearts perfused with a histone mixture support the hypothesis that this pathway may contribute to their harmful effects in polymicrobial sepsis.

The impact of TLR stimulation on ventricular myocyte contractility has been previously reported, underscoring TLRs’ critical role in modulating cardiac function [52]. Numerous reports on sepsis highlight the importance of TLR2, TLR3, and TLR4 in cardiac dysfunction and cardiomyopathy, although their precise roles remain incompletely understood. Our study explores histones as potential mediators of cardiomyopathy through interactions with TLRs. Consistent with findings in sepsis and other cardiomyopathy models, TLR2 KO and TLR3 KO are shown to be protective, whereas TLR4 KO exacerbates cardiac dysfunction and cardiomyopathy. For instance, Ma et al. reported that TLR2 blockade attenuated cardiac dysfunction and inhibited cardiac fibrosis in a chronic cardiomyopathy model induced by doxorubicin injection, while TLR4 blockade exacerbated cardiac dysfunction and fibrosis by amplifying myocardial inflammation [53]. Similarly, Mathur et al. found that TLR2 deficiency reduced CM inflammatory responses, whereas TLR4 deficiency had minimal impact [54]. In line, TLR2 KO mice showed better cardiac function and survival in a doxorubicin-induced cardiomyopathy model compared to Wt mice in another study [55]. Zhu et al. found that activating TLR4 offers a protective advantage to CMs, shielding them from stress-induced damage through MyD88- and NOS2-dependent pathways [56]. In terms of TLR3, in line with our results, Gao et al. showed that TLR3 deficiency attenuated cardiac dysfunction and increased survival outcomes following CLP-induced sepsis in mice [57].

Collectively, our research is consistent with the protective effects of TLR2 and TLR3 KO, and not TLR4 KO [53,54,56,57,58,59,60]. One limitation of our study is that we did not investigate the effect of each individual histone on cardiac function in TLR2 KO or TLR4 KO models in vivo. Further research is needed to determine whether specific histones, acting through individual TLRs, contribute to the observed mechanical and electrical changes in cardiac function. This could identify potential mechanistic targets unique to each histone subtype.

Furthermore, our grasp of how individual histones and their epigenetic modifications (acetylation, methylation, phosphorylation, ubiquitination, SUMOylation) influence biological responses remains limited. For example, the implications of converting arginine to citrulline in histones are not well understood. Previous studies using a neutralizing histone antibody (clone BWA3) that targets H2A and H4 [11,61], have shown reductions in the tissue-damaging effects of histones across various models, including polymicrobial sepsis [11,14], acute lung injury [19,62], and chemical-induced organ injury [23]. Furthermore, citrullination by peptidyl arginine deiminase 4 (PAD4) not only modifies histones but may also trigger the formation of autoantibodies, exacerbating autoimmune conditions such as rheumatoid arthritis and systemic lupus erythematosus [63,64,65]. Development of these auto antibodies may intensify the underlying autoimmune disease [66]. Understanding the intricate relationships between histone modifications and their biological outcomes continues to be a significant challenge, as evidenced by the variable effects of histone methylation on biological responses, highlighting the complex nature of histone-mediated signaling pathways.

Although the literature suggests that histones bind to TLRs releasing proinflammatory products [23,24,36,38,67], it is premature to define the specific functions of histones related to these interactions. With the presence of both native and chemically modified histones, there is still much to understand about their biological roles before we can effectively target them in acute conditions like sepsis, trauma, and acute lung injury. Most studies have focused on 28-day survival rates, leaving uncertainties about the immediate impacts on heart function in sepsis. Furthermore, clinical trials have not been structured to evaluate the developments following polymicrobial sepsis. Clinical trials from 2010 to 2021 investigated the role of endotoxin (LPS) and its interaction with TLR4 antagonists, such as eritoran and TAK-242 but yielded mixed results. Despite initial expectations, these studies failed to provide conclusive evidence of efficacy [58], leading the pharmaceutical industry to halt further research in septic patients. To improve outcomes, it is important to understand the molecular events unfolding during the early stages of sepsis in order to develop innovative therapies.

In conclusion, our findings underscore the intricate roles of histones and TLRs in inflammatory responses and cardiac dysfunction. Elucidating specific interactions between individual histones and TLRs is crucial for developing targeted therapies in acute inflammatory conditions such as sepsis. The need for new strategies to increase our knowledge of how histones adversely affect biological systems and cause cell and organ dysfunction, like cardiomyopathy, is a pressing problem. Specifically, focusing on H1 might offer an interesting prospect.

## 4. Materials and Methods

### 4.1. Animals

All animal procedures were performed in accordance with U.S. National Institutes of Health guidelines and were approved by the University of Michigan Committee on the Use and Care of Animals (protocol PRO00005749). Male age-matched C57BL/6 (wild type), TLR2-KO (TLR2^−/−^), TLR3-KO (TLR3^−/−^), and TLR4-KO (TLR4^−/−^) were purchased from the Jackson Laboratories (Bar Harbor, ME). All animals were housed under specific pathogen-free conditions with free access to food and water.

### 4.2. Whole Heart Perfusions, Functional LVP and Electrical (ECG) Measurements

Male and female adult C57BL/6 mice aged three to four months underwent treatment involving a peritoneal injection of 0.075 mL heparin followed by cervical dislocation. Subsequently, the hearts were rapidly excised and immersed in ice-cold PBS (pH 7.4). The aorta was cannulated on a Langendorff perfusion apparatus (AD Instruments, Colorado Springs, CO), and warm (37 °C), oxygenated Krebs–Henseleit Buffer (KHB) supplemented with 1.8 mM Ca^2+^ was perfused for 10–15 min to stabilize the heart. The left atrium was excised, and a water-filled balloon connected to a pressure transducer was inserted into the left ventricle to record LVP. Perfusate flow and balloon pressure were adjusted to achieve a diastolic pressure of around 8 mmHg and a systolic pressure of around 95 mmHg. Hearts unable to reach the target pressures with a maximal perfusion flow of 4 mL/min were discarded. Following an approximate 10 min stabilization period, the perfusion solution was switched to KHB (1.8 mM Ca^2+^) containing either the histone mix or individual histones. Surface electrodes were placed at L1 and L2 levels for ECG recordings, connected to an octal bioamp and a PowerLab unit. Raw tracings were filtered using 5 Hz high pass and 450 Hz low pass digital filters, and analysis was conducted using PowerLab software (version 7, AD Instruments, Boston, MA, USA). Each data point represents the average of at least 15 superimposed tracings. Semi-automated customized macros were employed to derive functional parameters for both LVP and ECG. Perfusate samples (2 mL) were collected at 0, 30, 60, 90, 120, and 150 min of histone perfusion, flash frozen, and stored at −80 °C for cytokine quantification. ECG measurements involved recordings for right versus left volume, with continuous perfusion of histones using Langendorff perfusions.

### 4.3. Protocols for Wt Mouse Heart Perfusion with Histone Mixture or Individual Histones

Wt or different TLRs KO Langendorff hearts including TLR2 KO, TLR3 KO, and TLR4 KO were perfused at 37 °C for various intervals of time with the histone mix. This histone mix was already tested in our earlier studies suggesting that the biological responses induced by histones in phagocytes were not due to the contamination with LPS [41,62]. We also investigated the effects of individual histones, including H1, H2A, H2B, H3, and H4, by perfusing them in Wt Langendorff hearts. Perfusion of hearts with the histone mix or individual histones continued for up to 150 min, with data collected at 30 min intervals, unless otherwise noted. Readouts included abnormal LV functions and defective ECG tracings, cytokine release, the release of LDH, and TUNEL staining.

### 4.4. Cytokine ELISA

Cytokines (IL-6 and TNF-α) levels were assessed in the collected perfusate samples using the R&D DuoSet ELISA kit (R&D Systems, Minneapolis, MN, USA) following the manufacturer’s guidelines.

### 4.5. LDH Cytotoxicity Assay and TUNEL Staining

An increase in LDH levels released from cells indicates damage to the cells and leakage resulting from membrane disruption. To assess the cardiac damage and cytotoxic effects induced by individual histones, we performed the LDH assay on the perfusate samples, following the guidelines provided by the manufacturer (Cayman Chemical, Ann Arbor, MI, USA), as previously demonstrated in our studies [36,41].

TUNEL staining on frozen heart tissue was applied according to the manufacturer’s recommendations, using the special kit (Promega, Madison, WI, USA) and performed as described in our recent publication [18]. Frozen sections of left ventricles were mounted in ProLong Gold Anti-Fade reagent containing DAPI (Life Technologies, Carlsbad, CA, USA). Images were acquired on a Nikon A-1 confocal system with Nikon Elements software (version 3.2, Nikon Instruments Inc., Tokyo, Japan). The average number of TUNEL-positive cells/field was determined in a blinded fashion from ≥10 fields over several sections per heart. Notably, while electrical and functional measurements were conducted with hearts being electrically paced, no pacing was employed for cytokine assessment and LDH/TUNEL staining.

### 4.6. Echo-Doppler Functional Studies on TLR2, TLR3, and TLR4 KO Mice

Echocardiograms were performed as we described in our previous studies [11,12,33,36,68] according to the recommendations of the American Society of Echocardiography. All echocardiograms were performed by a registered echocardiographer who was blinded to genotype. TLR2 KO mice (*n* ≥ 7), TLR4 KO mice (for each, *n* ≥ 7), and control Wt mice (*n* ≥ 7) were weighed and anesthetized with inhaled isoflurane. Imaging was performed using a Vevo 770 High-Resolution In Vivo Imaging System (Visualsonics Inc., Toronto, ON, Canada) equipped with an RMV-707 30 MHz RMV (Real-Time Visualization) (up to 45 MHz) scanhead. LV volumes were measured from the parasternal long-axis view at the level of the tips of the leaflets of the mitral valve at end-systole (VolS) and end-diastole (VolD) and used to calculate stroke volume (SV = VolD – VolS) and ejection fraction (EF % = endocardial SV/endocardial VolD × 100). Cardiac output (CO = SV × heart rate) was calculated from stroke volume and heart rate. Isovolemic relaxation time was determined from the apical 4-chamber view using pulse wave Doppler acquisition and was measured from the closure of the aortic valve to the onset of mitral inflow. Imaging was performed at 0, 10, 60, and 90 min and 24 h after histone (65 mg/kg body weight) injection (i.v.).

### 4.7. Reagents

Mixed calf thymus histones (purified, Type II-A) and histones H1 and H3 (purified from calf thymus) were obtained from Roche (Indianapolis, IN, USA). The remaining individual histones (H2A, H2B, and H4) were in recombinant form and were purchased from Cayman Chemical (Ann Arbor, MI, USA).

### 4.8. Statistical Analysis

For statistical analysis, GraphPad Prism version 10.0 (Graphpad, San Diego, CA, USA) was used. All values are expressed as mean ± SEM. Data sets were analyzed using one-way or two-way ANOVA, followed by Dunnett’s, Sidak, or Tukey’s multiple comparison tests as appropriate for each dataset. Survival curve comparisons were evaluated using the log-rank (Mantel-Cox) test with <0.05 considered significant.

## Figures and Tables

**Figure 1 ijms-25-08653-f001:**
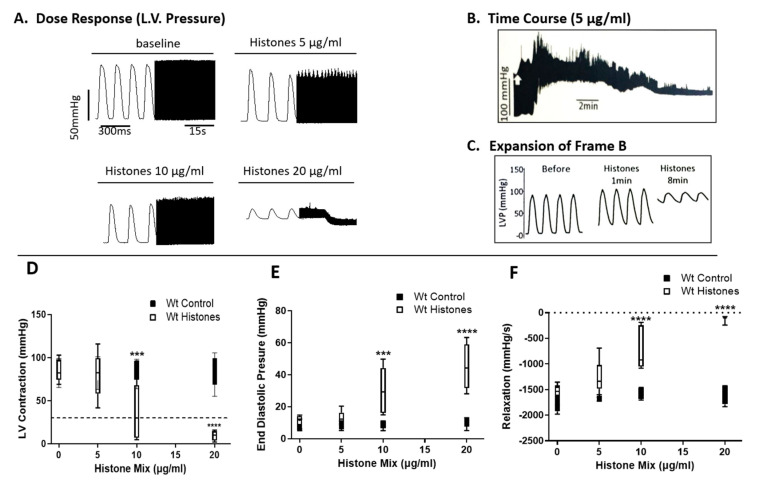
Functional abnormalities in Wt mouse hearts perfused ex vivo with histone mix. Wt mouse hearts, electrically paced, were perfused ex vivo with control buffer or histone mix (5, 10, and 20 µg/mL) for up to 20 min at 37 °C (**A**–**F**). Each condition involved 11 hearts. (**A**) Tracings for buffer and various histone mix concentrations showed progressive reduction and widening of left ventricular pressure (LVP) amplitudes. At 20 µg/mL, LVP amplitudes collapsed, inducing bradycardia. (**B**) Temporal progression after perfusing with a 5 µg/mL histone mix, revealing widening amplitudes and bradycardia within 1 min. (**C**) LVP collapsed by the 8th minute. (**D**–**F**) Displays LVP, end-diastolic pressures, and relaxation pressures, respectively, over 20 min. Dashed lines mark heart failure (≥80% pressure loss). Black bars represent buffer (Wt Control), while white bars represent histone mix (Wt Histones) at different doses. (**D**) Declining LVP with 10 µg/mL histone mix, and 20 µg/mL significantly reduces all pressures (**D**–**F**). (**F**) Relaxation pressures declined with histone mix, accentuated at 10 and 20 µg/mL concentrations. In (**D**–**F**), data are shown as max–min box plots and analyzed by two−way ANOVA followed by Sidak’s multiple comparison test. Significance is indicated as follows: *** *p* < 0.001 and **** *p* < 0.0001. The dashed line in (**D**) indicates the heart failure threshold, and in (**F**), shows 0 mmHg/s as the baseline.

**Figure 2 ijms-25-08653-f002:**
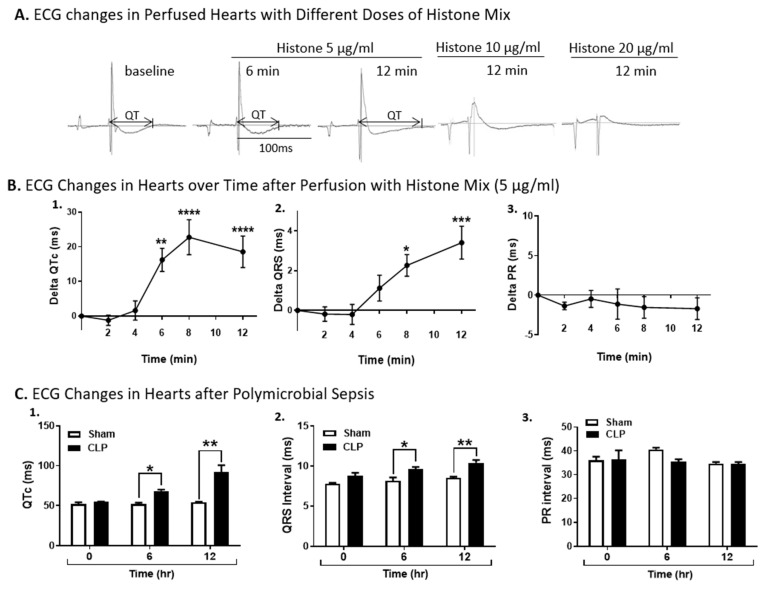
Electrocardiography (ECG) changes in Wt hearts perfused with histones or after induction of polymicrobial sepsis by cecal ligation and puncture (CLP). (**A**) Wt hearts (n ≥ 5) were perfused with histone mix (5, 10, and 20 µg/mL) for 12 min at 37 °C. By 12 min, there was a distinct QT interval widening, and at higher concentrations (10, 20 µg/mL), ECG tracings collapsed. (**B**) Hearts perfused with a 5 µg/mL histone mix for 12 min. (**B-1**) Increased QT interval peaking at 8 min, (**B-2**) steady increase in QRS interval, and (**B-3**) no change in PR interval. (**C**) ECG recordings before and after CLP in Wt mice (*n* = 5) at 6 and 12 h. Septic mice displayed significantly elevated QTc (Bazett’s formula) and QRS intervals at both time points, while PR interval remained unchanged. Sham controls showed no alterations. All values in (**B**,**C**) are expressed as means ± SEM. Data sets were analyzed by one-way ANOVA followed by Dunnett’s multiple comparison test for (**B**), and two-way ANOVA followed by Sidak’s multiple comparison test for (**C**). Significance is indicated as follows: * *p* < 0.05, ** *p* < 0.01, *** *p* < 0.001, and **** *p* < 0.0001.

**Figure 3 ijms-25-08653-f003:**
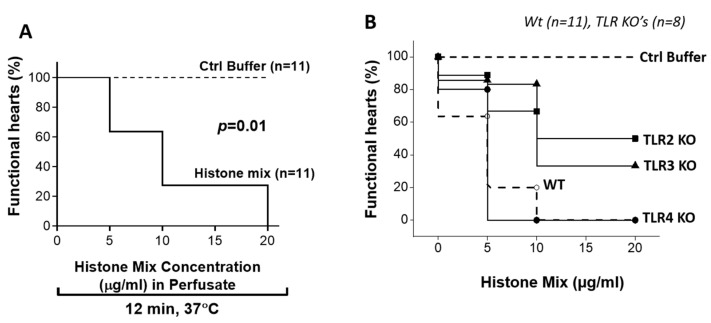
Ability of histone mix to cause cessation of beating mouse hearts as a function of histone concentration and effect of TLR absence. (**A**) Non-paced beating mouse hearts (Wt) were perfused with histone mix (5, 10, and 20 µg/mL) for 12 min at 37 °C. Cessation of beating hearts increased with histone concentration, reaching 100% at 20 µg/mL. (**B**) Comparison of Wt and TLRs KO hearts’ responses to 10 µg/mL histones. Wt hearts showed 20% beating, decreasing to 0% at 20 µg/mL. TLR2 KO exhibited better preservation, while TLR4 KO hearts ceased beating at all concentrations. LV dysfunction was observed in TLR2 KO (5 µg/mL), sparing 80% function. TLR3 KO provided partial protection. TLR4 absence offered no protection. Perfusing hearts with the control buffer (PBS) does not cause any functional defects, as shown by the dashed line. Comparisons of survival curves were analyzed using the log-rank (Mantel-Cox) test. *p* < 0.05 is considered as significant.

**Figure 4 ijms-25-08653-f004:**
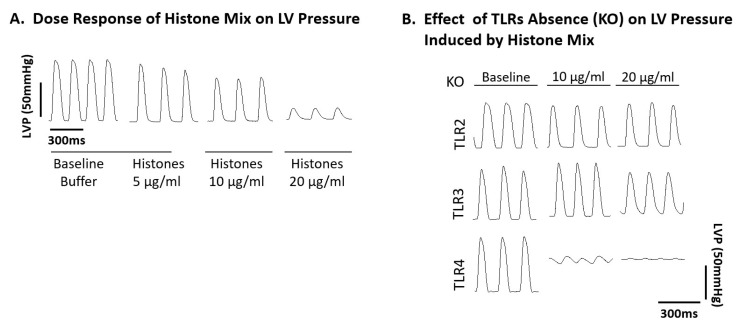
Functional abnormalities in Wt mouse hearts perfused with histone mix and effects of TLR absence on functional responses of mouse hearts to histone mix. Wt mouse hearts (n = 11) and hearts from TLR2, TLR3, and TLR4 KO mice (n ≥ 8 each) were electrically paced and perfused for 20 min at 37 °C with buffer (recorded as the baseline) or histone mix (5, 10, 20 µg/mL). (**A**) Wt hearts’ LVP amplitude reduction, collapsing at 20 µg/mL histones. (**B**) Illustrates TLR2, TLR3, and TLR4 KO baseline LVP (n = 8). TLR2 KO exhibited widened LV pressures with histones, TLR3 KO had a modest drop (20%) at 20 µg/mL, while TLR4 KO had virtually ablated LV pressures at both histone concentrations. TLR2 KO and TLR3 KO displayed protective effects with minimal changes or modest decline at higher histone doses. In contrast, TLR4 KO hearts were vulnerable, showing dysfunction even at lower histone concentrations.

**Figure 5 ijms-25-08653-f005:**
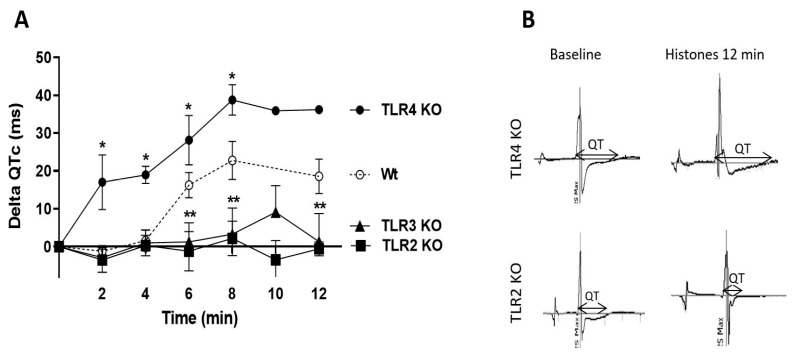
Effect of TLR absence on delta QTc prolongation in murine hearts perfused with histone mix. Wt paced hearts were perfused with buffer or histone mix (5, 10, 20 µg/mL) for 12 min, and ECG recordings were measured at 2 min intervals. (**A**) 5 µg/mL histone mix-induced QTc prolongation over 12 min in Wt hearts; TLR2, and TLR3 KO mice displayed minimal changes, while TLR4 KO mice exhibited exaggerated prolongation. (**B**) ECG patterns of TLR4 KO (worse) and TLR2 KO (protective). The values in 5A are expressed as means ± SEM. Data sets were analyzed by one-way ANOVA followed by Dunnett’s multiple comparison test. Significance between groups in each time point is indicated as * *p* < 0.05 and ** *p* < 0.01.

**Figure 6 ijms-25-08653-f006:**
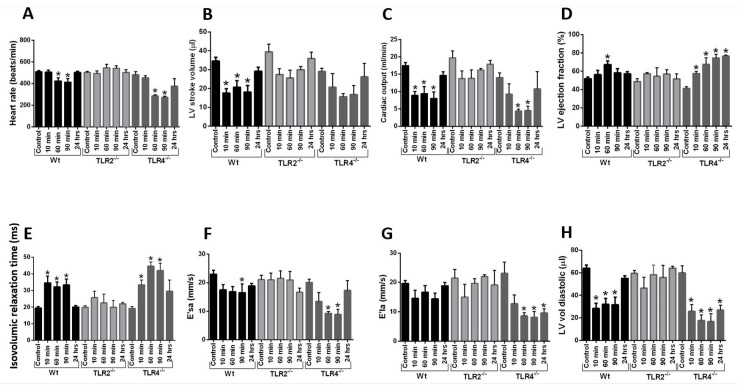
Echo−doppler parameters in Wt, TLR2 KO, and TLR4 KO hearts after intravenous infusion of histone mix. Effects of TLR2 or TLR4 KO on responses of mouse hearts after i.v. infusion of histone mix. Wt, TLR2 KO (protective), and TLR4 KO (least protective) mice were intravenously injected with histone mix (65 mg/kg). Echo−Doppler parameters were evaluated at various time points post−injection. (**A**) Significant heart rate decreases in Wt and TLR4 KO but not TLR2 KO groups 90 min post−injection, returning to baseline by 24 h. TLR4 KO exhibited a more intense drop compared to Wt. (**B**) Stroke volumes decreased significantly in Wt but not in TLR2 KO or TLR4 KO mice. (**C**) Cardiac output dropped significantly in Wt and TLR4 KO but not in TLR2 KO mice. (**D**) Left ventricular ejection fraction slightly increased in Wt and to a bigger extent in TLR4 KO after infusion. TLR2 KO showed no sustained change. (**E**) Isovolumic relaxation time increased significantly in Wt and TLR4 KO but not in TLR2 KO mice. (**F**–**H**) Other parameters displayed variable responses especially in TLR4 KO. Each bar represents Echo−Doppler measurements from 5 mice. Values are expressed as means ± SEM. Differences were assessed with one−way ANOVA, and group differences were determined using Tukey’s post–hoc test. Differences were considered significant if *p* < 0.05 as shown by one star.

**Figure 7 ijms-25-08653-f007:**
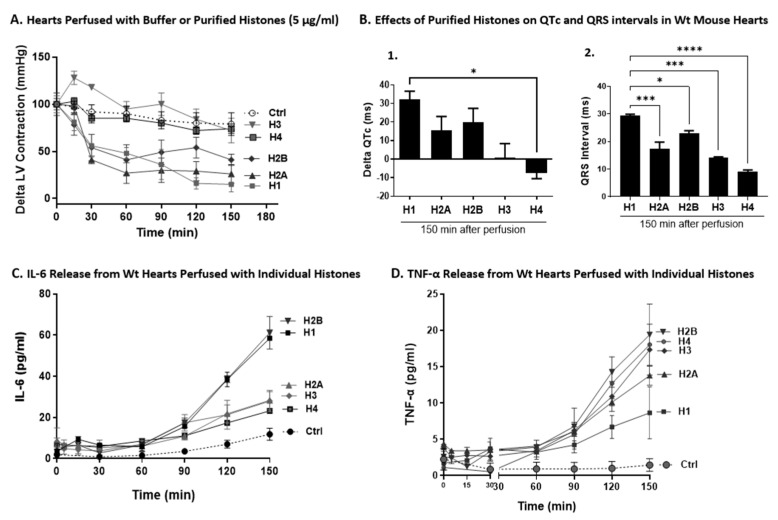
Effects of individual histones on LV function, ECG changes, and cytokine release on perfused hearts. (**A**) Significant reduction in LVP was observed with H1, followed by H2A and H2B. Minimal effects were observed with H4, while H3 slightly increased pressures. (**B**) QT interval pro–longation was noted with H1 (33 ms) and to a lesser extent with H2A and H2B (18–22 ms) but not with H3 and H4. (**B-1**) QT interval was more prolonged by perfusion with H1 compared to other individual histones. (**B-2**) QRS interval prolongation was caused by H1, H2A, and H2B. Overall, H1 emerges as the primary histone-inducing functional and electrical dysfunction. (**C**) Cytokine release was highest for IL-6 with H1 and H2B, followed by H2A, H3, and H4. (**D**) TNF-α responses were most pronounced with H2B, followed by H4, H3, H2A, and H1. Statistical analyses included one–way ANOVA with Tukey’s post–hoc analysis for comparisons among histones in (**A**) and with H1 in (**B**). Significant differences in (**A**) are as follows: Control vs. H1 *p* < 0.05), Control vs. H2A (*p* < 0.05), H1 vs. H3 (*p* < 0.01), H2A vs. H3 (*p* < 0.01), and H2B vs. H3 (*p* < 0.05). In (**A**), these *p*–values are not shown due to space constraints. In (**B**), * *p* < 0.05. *** *p* < 0.001, **** *p* < 0.0001.

**Figure 8 ijms-25-08653-f008:**
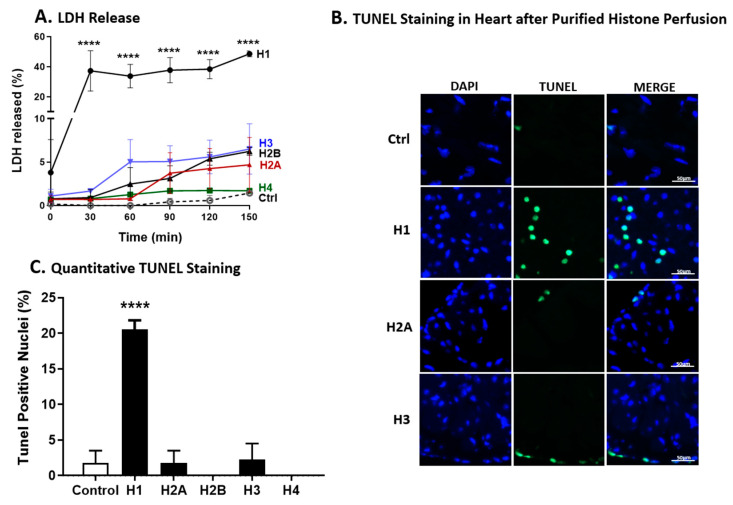
LDH Release and TUNEL Staining of Wt Hearts Atria after Perfusion with Individual Histones. Wt mouse hearts were perfused with individual histones (5 µg/mL) or buffer at 37 °C for specified durations. (**A**) Within 30 to 150 min, H1 induced approximately 40% LDH release, primarily in the initial 30 min while other histones caused only 2–5% release. (**B**) TUNEL staining demonstrates that H1 induced maximal apoptosis (around 20%). Blue: DAPI staining; Green: TUNEL staining. (**C**) Other histones showed minimal staining. H2B and H4 exhibited no staining. Scale bars = 50 µm. Values are expressed as means ± SEM. Differences were assessed with one-way ANOVA, and group differences were determined with Tukey’s procedure. Differences were considered significant if *p* < 0.05. In (**A**,**C**), **** *p* < 0.0001.

## Data Availability

Data is contained within the article.

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
