# Peer review of "The Impact of Extracellular Histones and Absence of Toll-like Receptors on Cardiac Functional and Electrical Disturbances in Mouse Hearts"

_ijms, 2024, doi:10.3390/ijms25168653_

Round 1

Reviewer 1 Report

Comments and Suggestions for Authors

In this manuscript, the authors demonstrated the extracellular histone subtypes and the toll-like receptors (TLRs) in cardiac dysfunctions using Langendorff perfusion and knockout mice models. This research is designed step-by-step and well-structured.

Although this study is interesting, several issues remain to be addressed:

-          Wt mouse heart perfused with 5μg/ml H2B for 150min displayed the highest IL-6 and ENF release (Figure 7C,D) but showed ZERO apoptosis (TUNEL staining, Figure 8B). Please clarify the reason.

-          The authors demonstrated that TLR2 KO showed protection while TLR4 KO showed worse effect. And the different histone subtypes displayed respective effects on Wt mouse. Why not apply these histones on TLR2 and TLR4 KO mouse models?

-          The authors mentioned QT interval was recorded in the text, while QTc was shown in the figures, if QTc used, please clarify which formula was used to calculate the corrected QT interval (QTc).

-          Indicate the statistical method used and the groups involved in all figures.

-          Amend all ‘ug’ to ‘μg’.

-          Figure 1: Why not include 15μg/ml Histone group as it showed in Figure 1D-F? if not, please exclude this point from the figure. Indicate the data presentation in Figure 1, mean ± SD or mean ± SEM? Indicate the line in Figure 1D and 1F.

-          Figure 2: Indicate the prefusion time in two higher Histone groups (10 and 20 μg/ml) in frame A.

-          Figure 4: Use the same ‘ms’ scale bar in frame A and B.

-          Figure 5A: Why 10min group in Wt is missing?

-          Figure 8B: Include scale bar in the immunofluorescence staining images.

-          Line 620: Amend ‘μg/gm’ to ‘μg/mg’ or ‘mg/kg’.

Author Response

Reviewer 1

1.     Wt mouse heart perfused with 5µg/ml H2B for 150min displayed the highest IL-6 and TNF release (Figure 7C,D) but showed ZERO apoptosis (TUNEL staining, Figure 8B). Please clarify the reason.

Response: Thank you for your insightful comments. We chose a concentration of 5 µg/ml for our investigation into the effects of histones on beating hearts because this level led to measurable heart dysfunction, including arrhythmia and contractile dysfunction, without causing immediate failure. This allowed the hearts to remain viable for more than an hour. At this concentration, we also documented the release of TNF-α and IL-6, as evidenced by Figures 7C and 7D. As you noted, there was no apoptosis observed by TUNEL staining, as shown in Figure 8B. When we increased the histone concentration to 10 µg/ml or more, the hearts quickly progressed to failure, accompanied by positive TUNEL staining, indicating apoptosis. The absence of apoptosis at the 5 µg/ml concentration suggests a threshold effect, where the lower concentration triggers inflammatory cytokine release but does not initiate the cell death pathways detectable by TUNEL staining. At higher concentrations, it is plausible that factors beyond cytokines contribute to the observed cell death as the role of TNF-α in cardiac disease seems to be context dependent (PMID: 38059906). In our model, these cytokines do not seem to be driving cell death, although a more prominent role cannot be ruled out in chronic disease. We have explained this response to the reviewer's point in the Results section, now detailed in lines 421-427.

2.     The authors demonstrated that TLR2 KO showed protection while TLR4 KO showed worse effect. And the different histone subtypes displayed respective effects on Wt mouse. Why not apply these histones on TLR2 and TLR4 KO mouse models?

Response: We appreciate the reviewer's suggestion. As shown in Figures 1 through 5, we conducted dose-response experiments on various physiological cardiac responses in an ex vivo model to understand the effects of different histone subtypes (H1, H2A, H2B, H3, and H4). Conducting in vivo studies in mouse models would necessitate comprehensive Echo-Doppler studies, which are prohibitively expensive and beyond the scope and budget of our current study. For instance, the cost of recombinant proteins for each histone subtype, if used in vivo (65 mg/kg body weight of mouse), is estimated between $500-$1000 per mouse (weighing around 18-25 grams). To achieve sufficient statistical power, we would need at least 5-6 mice per group, multiplied by 5 histone subtypes, resulting in at least 75 experimental readouts for 3 strains of mice (Wt, TLR2 KO and TLR4 KO). Additionally, each mouse would require echocardiographic assessments at 5 different time points (baseline and 4 post-injection), each costing at least $100 per time point, further increasing the total cost to several hundred thousand dollars. This expense is beyond our financial capabilities and the goals of our current research project. Respecting the reviewer's comment, we have acknowledged this limitation in our study and highlighted the need for future research to determine which histone subtypes have the most significant impact on cardiac dysfunction in TLR2 and TLR4 KO mice in vivo (Discussion section, lines 522-527). This important next step would fully elucidate the roles of these histones in cardiac pathology. We have addressed this point in the closing remarks of the Discussion section (line 563).

3.     The authors mentioned QT interval was recorded in the text, while QTc was shown in the figures, if QTc used, please clarify which formula was used to calculate the corrected QT interval (QTc).

Response: We used Bazett´s formula for QTc correction. We added this to the text (line 203). However, as demonstrated recently (PMID: 38261726), in mice the QT interval is the same regardless of whether or not it is corrected. This contention is in agreement with a number of reports. 

4.     Indicate the statistical method used and the groups involved in all figures.

Response: We addressed this to all figure legends as requested. We also added the missed section of ‘Statistical Analysis’ in the last section of Materials and Methods (section 4.8)

5.     Amend all 'ug' to 'µg'.

Response: We changed “µg” to “µg” throughout the manuscript as requested.

6.     Figure 1: Why not include 15µg/ml Histone group as it showed in Figure 1D-F? if not, please exclude this point from the figure. Indicate the data presentation in Figure 1, mean ± SD or mean ± SEM? Indicate the line in Figure 1D and 1F.

Response: We designed our experiment to include a baseline and three histone concentrations: 5, 10, and 20 µg/ml, as shown in Figures 1-4. The reviewer correctly points out the inclusion of the 15 µg/ml label on the axis of our figures, which was not part of our experimental design. We acknowledge this oversight. The 15 µg/ml label appeared due to the axis scaling, which automatically included intervals of 5 µg/ml to maintain consistency and clarity in data representation. To avoid any confusion, we have revised the figure legends for Figures 1-4 to clearly state that the studied concentrations of histones were 5, 10, and 20 µg/ml. The legends now read: " hearts were perfused with histone mix (5, 10 and 20 µg/ml)". Additionally, we have clarified the data presentation in Figure 1, specifying that the values are presented as max-min box plots. We appreciate the reviewer’s suggestion, which led us to correct the number of stars indicating the p-value according to the two-way ANOVA test. We also indicated the dashed lines in the legends of Figures 1D and 1F, ensuring accurate interpretation. The dashed line in D indicates the heart failure threshold, and in F, shows 0 mmHg/s as the baseline.

7.     Figure 2: Indicate the perfusion time in two higher Histone groups (10 and 20 µg/ml) in frame A.

Response: As already stated in the legend of Figure 2, Wt hearts (n ≥ 5) were perfused with histone mix (5, 10, 20 µg/ml) for 12 min at 37°C. By 12 min, there was a distinct QT interval widening, and at higher concentrations (10, 20 µg/ml), ECG tracings collapsed. We added “12 min” to those two higher histone groups (10, 20 µg/ml) in frame A as requested.

8.     Figure 4: Use the same 'ms' scale bar in frame A and B.

Response: Done. Thank you for the comment.

9.     Figure 5A: Why 10min group in Wt is missing?

Response: Delta QTc at 10 min was not measured in the control Wt because the data collection for that specific time point was inadvertently missed during the experiment. We relied on the extrapolation between 8 and 12 min.

10.  Figure 8B: Include scale bar in the immunofluorescence staining images 

Response: Thank you for your comment. The immunofluorescence staining images in Figure 8B were captured at 40x magnification, so we included this information in the legend. We appreciate the reviewer's attention to detail.

11.  Line 620: Amend 'µg/gm' to 'µg/mg' or 'mg/kg'.

Response: We corrected the unit and made it consistent throughout the text as “mg/kg” body weight. Thank you.

Reviewer 2 Report

Comments and Suggestions for Authors

The manuscript needs corrections. Here are some remarks:

1.      The introduction needs better explanation why investigation of TLR were chosen to investigate and the effect of histones on them.

2.      The description of figures in the text should be marked as (Fig. 1A, Fig 1B and so on) instead (frame A, B…). The same remark valid for all figures.

3.      Fig. 2 is missing explanations of statistical significance. Fig. 2A title is missing “of”. All abbreviations used in Fig. 2 should be explained in the legend.

4.       (The Figure 3) in the title of Section 2.3. should be eliminated. Similar remarks are about all titles of sections.

5.      Fig. 3 is talking about the TLR in Wt or TLR4 KO hearts, however, the levels of TLR are not shown. There are many ways to do it. The proof that the level of TLR in Wt and TLR4 KO hearts are different should be shown. Similar remarks are about the other types of TLR KO mice in comparison to the Wt.

6.      Fig.3B – the control is missing.

7.      Fig. 4 – the control is missing.

8.      Fig. 6. The statistical significance is not explained in the legend. The parameters mentioned in Fig. 6, as well as in other figures needs better explanation in the text, i.e. what their increase or decrease means and how the reader should interpret those parameters.

9.      Fig.7 – the statistical significance analysis and its description is missing. Which type of TNF was analyzed?

10.  Fig.8 A - the statistical analysis is missing. Fig. 8 legend - the description of statistical parameters is missing. Scale bars and their values exploration is also missing.

11.  Discussion needs better explanation what is so special about H1 and its effects compared to the other histones. The conclusions should be more concrete.

12. The statistical part in methods is missing as well as TLR evaluation.

13.      The bioethics permission to work with animals is also missing.

Comments on the Quality of English Language

The English language is quite good.

Author Response

Reviewer 2

  1. The introduction needs better explanation why investigation of TLR were chosen to investigate and the effect of histones on them.

Response:  The literature contains several reports where the absence (knockout) of TLR2, TLR3, and TLR4 has been used to define the biological roles of these TLRs. Research highlights the potential of pharmacological TLR signaling regulation for treating diseases like sepsis. However, very few have progressed to clinical trials or gained approval, with none proving effective thus far (PMID: 28769820). Even though TLR4 was considered to be an important TLR, its blockade in a large international clinical trial of sepsis patients failed to show protection in humans (PMID: 23512062). The use of KO mice is gradually providing useful data, but we still poorly understand the role of TLR receptors in sepsis. As we explained through our Introduction, release of histones to extracellular space (mainly from neutrophils as a native response to the pathogens) is an event that happens during sepsis. As stated in our Introduction (line 105-111), TLRs, especially TLR2 and TLR4, have been suggested as receptors for extracellular histones (PMID: 19855397, PMID: 21784973). We previously demonstrated that TLR3 and TLR9 could be also potential receptors for histones, as shown by echocardiographic studies indicating the protective effect of their knockout (PMID: 30364002). Both we and other researchers have shown that extracellular histones act as DAMPs (Damage-Associated Molecular Patterns) released during NET (neutrophil extracellular trap) formation, which was the basis for our study design. There has been increasing evidence over the past few years that responses developing during sepsis in mice are linked to TLR activation, particularly TLR2, TLR3, and TLR4. However, the precise mechanisms of how TLRs contribute to these responses are not fully understood. Numerous papers suggest that some TLRs enhance proinflammatory responses during sepsis. However, there is no consensus on the exact roles of individual TLRs in inflammatory disorders such as polymicrobial sepsis. This is a critical issue, as evidenced by the failure of a large international study to show that blocking TLR activity alleviates the destructive effects of sepsis in humans. Clearly, there is much to learn about the pathophysiology of sepsis in humans and the best targets for therapeutic intervention. Understanding the roles of histones and TLRs in sepsis is urgent and essential for developing effective treatments to alleviate the harmful effects of this condition. Therefore, our study aimed to investigate the effects of extracellular histones on TLR2 and TLR4, building on the existing evidence and addressing these crucial gaps in knowledge.

  1. The description of figures in the text should be marked as (Fig. 1A, Fig 1B and so on} instead (frame A, B... ). The same remark valid for all figures.

Response:  All figure/frame labels have been changed to Fig. 1A, Fig. 1B and so on for simplicity.

  1. 2 is missing explanations of statistical significance. Fig. 2A title is missing "of'. All abbreviations used in Fig. 2 should be explained in the legend.

Response:  We have used the traditional statistical significance that employs the p scale, with statistical significance occurring when p values are <0.05. We added an explanation for the statistical significance in the figure legends of all figures. We have added “of” to the title of Fig. 2A and added abbreviations in the figure legend of Fig. 2 as requested.          

  1. (The Figure 3) in the title of Section 2.3. should be eliminated. Similar remarks are about all titles of sections.

Response: We have removed the figure numbers from the section titles.

  1. 3 is talking about the TLR in Wt or TLR4 KO hearts, however, the levels of TLR are not shown. There are many ways to do it. The proof that the level of TLR in Wt and TLR4 KO hearts are different should be shown. Similar remarks are about the other types of TLR KO mice in comparison to the Wt.

Response: Figure 3, panel A, shows heart function after histone perfusion only in Wt hearts. Panel B repeats the experiment including TLR2 KO, TLR3 KO, and TLR4 KO, comparing them with the Wt responses. We now added the absence of TLR in KO mice in the first paragraph of the Methods section (4.1) in our revised manuscript to avoid confusion. Figure 3 does not address the levels of TLRs in Wt hearts. To clarify, the question seems to ask whether TLR4 levels in Wt and KO hearts are different. We apologize if we misunderstood the reviewer’s comment.

  1. 3B - the control is missing.

Response: The control is represented by the baseline value, where no histone perfusion was performed and only buffer (PBS) was perfused which is the same as Fig. 3A. For clarity, we have added the control to the figure as well as the legend.

  1. 4 - the control is missing.

Response: The control is control buffer (PBS), where no histone perfusion was performed and only PBS buffer was perfused. This was indicated as the baseline in both frame A (for the hearts from wild type mice) and frame B (for the hearts from KO mice). For clarity, we have added this information to the figure legend.

  1. 6. The statistical significance is not explained in the legend. The parameters mentioned in Fig. 6, as well as in other figures needs better explanation in the text, i.e. what their increase or decrease means and how the reader should interpret those parameters.

Response: The areas marked with stars indicate significant reductions or increases in responses. We have completed explanations regarding these changes in the text (lines 344-367) and also added this information to the figure legend of Fig. 6, to help readers interpret the parameters more effectively.

  1. 7 - the statistical significance analysis and its description is missing. Which type of TNF was analyzed?

Response: We used TNF-α. We acknowledge that “TNF” is often used interchangeably with “TNF-α” in scientific literature. We have now specified “TNF-α” in our text and figure to avoid any confusion. Additionally, we have incorporated the statistical significance analysis into our text, as suggested.

  1. 8 A- the statistical analysis is missing. Fig. 8 legend - the description of statistical parameters is missing. Scale bars and their values exploration is also missing.

Response: Thank you for your feedback. We’ve updated Fig. 8A and its legend with the statistical analysis, detailed statistical parameters, and added the 40x magnification scale. We appreciate your insightful comments.

  1. Discussion needs better explanation what is so special about H1 and its effects compared to the other histones. The conclusions should be more concrete.

Response: Thank you for bringing this to our attention. We have revised the Discussion and Conclusions sections to better explain the unique and potent effects of histone H1 in comparison to other histone subtypes. Our previous studies in vitro and in vivo have shown that H1 has significant harmful effects, including inducing apoptosis, cell swelling and LDH release in acute lung injury. In comparison to other histone subtypes (H2A, H2B, H3, and H4), H1 has been found to cause the most robust harmful effects. Our study emphasizes the effects of H1 on cardiomyocytes, which we found to be a significant contributor to cardiac dysfunction. While our research has provided valuable insights, we acknowledge the need for further studies to determine the most effective target for treating heart dysfunction in humans. These future studies should investigate the effects of individual histones on heart function in vivo, as well as their impact on other organs during sepsis.

  1. The statistical part in methods is missing as well as TLR evaluation.

Response: As for the statistical part in our methods, we apologize if it was not clearly outlined. We have now updated the Methods section to include a detailed description of the statistical analysis used in our study. We would like to clarify that we did not evaluate TLR in our study. The TLR KO mice (TLR2, TLR3, TLR4 KO) were purchased from Jackson Laboratories. We stated this missed information as the first paragraph of our Methods section (4.1).

  1. The bioethics permission to work with animals is also missing.

Response: The Institutional Review Board statement for animal use approval is now added before the Acknowledgements.  We have also included it at the beginning of the Methods section (lines 678-681 and lines 568-570).

Round 2

Reviewer 1 Report

Comments and Suggestions for Authors

The manuscript was improved after addressing previous concerns.

Minor comments:

1. Move the figure 2 before line 195.

2. Add a scale bar in Figure 8B.

Author Response

Reviewer 1

  1. Move the Figure 2 before line 195.

Response: We have moved Figure 2 before line 195 as requested.

  1. Add a scale bar in Figure 8B

Response: We have added a scale bar to Figure 8B as requested in the merged images. We have also updated the figure legend to include "Scale bars = 50 µm" instead of “Images were captured at 40x magnification”.

Reviewer 2 Report

Comments and Suggestions for Authors

The manuscript has been corrected according to the reviewers suggestions. Minor remark - Fig. 2 legend in journals' template needs correction.

Author Response

Reviewer 2

  1. The manuscript has been corrected according to the reviewers suggestions.

Minor remark – Fig. 2 legend in journals’ template needs correction.

Response: We have revised the legend for Figure 2 in accordance with the journal's template.